# Characteristics and Predictors of Progression Interstitial Lung Disease in Rheumatoid Arthritis Compared with Other Autoimmune Disease: A Retrospective Cohort Study

**DOI:** 10.3390/diagnostics11101794

**Published:** 2021-09-28

**Authors:** Natalia Mena-Vázquez, Marta Rojas-Gimenez, Carmen María Romero-Barco, Sara Manrique-Arija, Ana Hidalgo Conde, Rocío Arnedo Díez de los Ríos, Eva Cabrera César, Rafaela Ortega-Castro, Francisco Espildora, María Carmen Aguilar-Hurtado, Isabel Añón-Oñate, Lorena Pérez-Albaladejo, Manuel Abarca-Costalago, Inmaculada Ureña-Garnica, Maria Luisa Velloso-Feijoo, Rocio Redondo-Rodriguez, Antonio Fernández-Nebro

**Affiliations:** 1Instituto de Investigación Biomédica de Málaga (IBIMA), 29010 Málaga, Spain; menchu01@hotmail.com (C.M.R.-B.); sarama_82@hotmail.com (S.M.-A.); inuregar@gmail.com (I.U.-G.); rocioredondo91@hotmail.com (R.R.-R.); afnebro@gmail.com (A.F.-N.); 2UGC de Reumatología, Hospital Regional Universitario de Málaga, 29009 Málaga, Spain; 3Instituto Maimónides de Investigación Biomédica de Córdoba (IMIBIC), 14004 Córdoba, Spain; rojasgimenezm@gmail.com (M.R.-G.); orcam84@hotmail.com (R.O.-C.); 4UGC de Reumatología, Hospital Universitario Reina Sofía de Córdoba, 14004 Córdoba, Spain; 5UGC de Reumatología, Hospital Clínico Universitario Virgen de la Victoria, 29010 Málaga, Spain; 6Servicio de Medicina Interna, Hospital Universitario Virgen de la Victoria, 29010 Málaga, Spain; ahidalgoconde@gmail.com (A.H.C.); rocioardiez@gmail.com (R.A.D.d.l.R.); maabco@gmail.com (M.A.-C.); 7UGC Neumología, Hospital Universitario Virgen de la Victoria, 29010 Málaga, Spain; evacabreracesar@gmail.com; 8UGC de Neumología, Hospital Regional Universitario de Málaga, 29009 Málaga, Spain; fespildorahernandez@gmail.com; 9UGC de Radiodiagnóstico, Hospital Regional Universitario de Málaga, 29009 Málaga, Spain; maguh007@gmail.com; 10Hospital Universitario de Jaén, 23007 Jaén, Spain; isaanononate@gmail.com; 11Hospital Universitario Virgen de las Nieves, 18170 Granada, Spain; lorenaperezalba@gmail.com; 12Hospital Universitario Virgen de Valme, 41014 Sevilla, Spain; mlvelloso@hotmail.com; 13Departamento de Medicina, Universidad de Málaga, 29010 Málaga, Spain

**Keywords:** rheumatoid arthritis, systemic autoimmune disease, interstitial lung disease, prognosis

## Abstract

Objectives: To describe the characteristics and progression of interstitial lung disease in patients with associated systemic autoimmune disease (ILD-SAI) and to identify factors associated with progression and mortality. Patients and methods: We performed a multicenter, retrospective, observational study of patients with ILD-SAI followed between 2015 and 2020. We collected clinical data and performed pulmonary function testing and high-resolution computed tomography at diagnosis and at the final visit. The main outcome measure at the end of follow-up was forced vital capacity (FVC) >10% or diffusing capacity of the lungs for carbon monoxide >15% and radiological progression or death. Cox regression analysis was performed to identify factors associated with worsening of ILD. Results: We included 204 patients with ILD-SAI: 123 (60.3%) had rheumatoid arthritis (RA), 58 had (28.4%) systemic sclerosis, and 23 (11.3%) had inflammatory myopathy. After a median (IQR) period of 56 (29.8–93.3) months, lung disease had stabilized in 98 patients (48%), improved in 33 (16.1%), and worsened in 44 (21.5%). A total of 29 patients (14.2%) died. Progression and hospitalization were more frequent in patients with RA (*p* = 0.010). The multivariate analysis showed the independent predictors for worsening of ILD-SAI to be RA (HR, 1.9 [95% CI, 1.3–2.7]), usual interstitial pneumonia pattern (HR, 1.7 [95% CI, 1.0–2.9]), FVC (%) (HR, 2.3 [95% CI, 1.4–3.9]), and smoking (HR, 2.7 [95%CI, 1.6–4.7]). Conclusion: Disease stabilizes or improves after a median of 5 years in more than half of patients with ILD-SAI, although more than one-third die. Data on subgroups and risk factors could help us to predict poorer outcomes.

## 1. Introduction

Interstitial lung disease (ILD) is a common condition in patients with systemic autoimmune diseases (SAIs) that is characterized by increased morbidity and mortality [1]. The SAIs most associated with ILD (ILD-SAI) include systemic sclerosis (SS), rheumatoid arthritis (RA), and inflammatory myopathy (IM). These conditions are reported to affect up to 70% of patients with ILD [2]. The most frequent subtype of ILD in high-resolution computed tomography (HRCT) and histopathology is nonspecific interstitial pneumonia (NSIP), except in RA, where the most frequent subtype is usual interstitial pneumonia (UIP) [3].

ILD is the main cause of death in patients with SS and IM [3,4,5], whereas in RA, it is the second cause of death after cardiovascular disease [1]. Furthermore, the percentage of patients who experience progression of ILD in all these diseases is variable [4,5,6]. Some studies have tried to identify factors that can help us to predict a poorer prognosis and/or greater mortality in patients with ILD-SAI. Poorer prognosis of ILD-SAI has been associated with demographic factors, such as advanced age [7,8,9,10,11,12], male sex [8,10,13,14], and history of smoking [2,15]. Similarly, reduced forced vital capacity (FVC) and diffusing capacity of the lung for carbon monoxide (DLCO) [15,16,17] and the UIP radiological pattern in HRCT have also been associated with marked progression and mortality in the different types of ILD-SAI [5,12,18,19,20,21,22,23,24,25,26]. The most common drugs for treatment of this condition include corticosteroids, cyclophosphamide, mycophenolate mofetil, and azathioprine [2]. Similarly, antifibrotic agents, such as nintedanib, could have a beneficial effect on the lungs of affected patients, as shown by the studies SENSCIS [27] and INBUILD [28]. In recent years, treatment with the immunosuppressants rituximab and abatacept has been reported to be safe and effective in ILD-SAI [29,30,31,32,33,34,35].

Knowledge of the factors associated with progression and mortality of ILD-SAI is important for identifying patients who may require more intensive therapy or earlier referral for evaluation of lung transplant. However, it is also interesting to identify epidemiologic and clinical differences, as well as differences in how progress of the various types of ILD-SAI affects the lungs, since this will provide a better picture of the progress of each group of patients and make it possible to identify more susceptible groups. However, few studies have compared the epidemiological, clinical, and progression-related characteristics of ILD-SAI using HRCT and pulmonary function testing (PFT). Therefore, the objectives of our study were as follows: (1) to describe the clinical and epidemiological characteristics of patients with ILD-SAI; (2) to compare the progression of lung disease and mortality in the various types of ILD-SAI; and (3) to identify factors associated with progression and mortality in patients with ILD-SAI.

## 2. Materials and Methods

### 2.1. Design

We performed a multicenter retrospective observational study of a cohort of patients with ILD-SAI from 6 teaching hospitals in Andalusia, Spain. The study was approved by the Research Ethics Committee of Hospital Regional Universitario de Málaga (HRUM), Malaga, Spain (Código 2017-N-19). All the participants gave their written informed consent for their personal information to be included in the database. 

### 2.2. Study Population

The study population comprised all patients with ILD-SAI in follow-up at the rheumatology clinic between January 2015 and December 2020. ILD was confirmed by PFT and HRCT or lung biopsy. The eligibility criteria were as follows: age ≥ 18 years, RA classified according to the ACR/EULAR 2010 criteria [36], SS according to the ACR/EULAR 2013 criteria, and dermatomyositis and polymyositis (IM) according to the criteria of Bohan and Peter [37,38], as applicable. Patients with an inflammatory or rheumatic disease other than RA, SS, or IM were excluded (except for secondary Sjögren syndrome).

### 2.3. Protocol

Patients were seen every 3–6 months in the rheumatology clinic and every 6–12 months in the pulmonology clinic. They also systematically underwent HRCT and PFT at diagnosis of ILD (V0), at any other visit if they presented symptoms of respiratory impairment or if their physician considered these investigations necessary, and during 2019–2020 to determine lung progression data on the inclusion date (Vf). All HRCT scans were based on axial slices measuring 1.5 or 2 mm in thickness at 1-cm intervals along the thorax and reconstructed using a high-spatial-frequency algorithm, with acquisition of 20–25 slices per patient. The radiological evaluation was centralized at HRUM and performed blind and independently by 2 experts in pulmonary radiology. Discrepancies between the readings were resolved by mutual agreement. Data were collected at V0 and Vf.

### 2.4. Working Definitions and Variables

The main variable was “Course of ILD at the end of follow-up (Vf)” with respect to the following: (1) improvement (i.e., improvement in FVC ≥ 10% or DLCO ≥ 15% and no radiological progression); (2) nonprogression (stabilization or improvement in FVC ≤10% or DLCO <15% and no radiological progression); (3) progression (worsening of FVC > 10% or DLCO > 15% and radiological progression); or (4) death [31]. Radiological progression was defined as a ≥20% increase in the presence and extension of ground-glass opacities, reticulation, honeycombing, diminished attenuation, centrilobular nodules, other nodules, emphysema, or consolidation compared with the HRCT images acquired at inclusion. 

The different ILD patterns were defined according to the lung biopsy or HRCT based on the standard criteria of the American Thoracic Society/European Respiratory Society International Multidisciplinary Consensus Classification of the Idiopathic Interstitial Pneumonias [39]. The 3 patterns defined were as follows: nonspecific interstitial pneumonia (NSIP), usual interstitial pneumonia (UIP), and other (bronchiolitis obliterans [BO], organizing pneumonia [OP], lymphoid pneumonitis, and mixed patterns). PFT included full spirometry expressed as percent predicted and corrected for age, sex, and height. An FVC value < 80% predicted was considered abnormal. DLCO was evaluated using the single-breath technique (DLCO-SB) and was considered abnormal at DLCO-SB < 80%. 

Other variables included time with symptoms, diagnostic delay, smoking history (current or previous), and co-occurrence of Sjögren syndrome. We recorded the infections in the clinical history, as well as the clinical event hospitalization and its causes. We also recorded laboratory variables, such as autoantibodies, rheumatoid factor (RF, reference, 20 U/mL; high titer > 60 U/mL), anticitrullinated peptide antibody (ACPA) (reference, 10 U/mL; high values > 340 U/mL), antinuclear antibody (ANA), anti-U1RNP (MCTD), anti-Scl70, anti-RNA polymerase III, anti-PM-Scl (PM-Scl overlap), anti-Ro 52 kDa, anti-Ro 60 kDa, anti-La, anti-aminoacyl-tRNA synthetase, anti-Mi-2, anti-SRP, anti-TIF1, anti-NXP-2/MJ, anti-MDA5 (CADM), anti-HMGCR, and anti-SAE. We recorded treatment with conventional synthetic disease-modifying antirheumatic drugs (csDMARDs), targeted synthetic DMARDs (tsDMARDs), biologic DMARDs (bDMARDs), immunosuppressants, and antifibrotic drugs, as well as corticosteroids. 

### 2.5. Statistical Analysis

A descriptive analysis was made of the clinical, epidemiological, and autoimmune characteristics and the treatment they received. Qualitative variables were expressed as absolute number and percentage and qualitative variables as mean and standard deviation (SD) or median and interquartile range (IQR), depending on the normality of the distribution, as assessed using the Kolmogorov-Smirnov test. A χ2 or ANOVA test or Kruskal-Wallis test was performed (as applicable) to compare the main characteristics between the 3 groups of patients: (1) patients with ILD-RA; (2) patients with ILD-SS; and (3) patients with ILD-IM. The bivariate analysis was performed using a paired *t* test or Wilcoxon test, as applicable, between V0 and the end of follow-up. Kaplan-Meier curves and the log-rank test were used to estimate the survival of patients with ILD-SAI and to compare survival between the 3 groups of patients. Survival was measured from diagnosis of ILD to the end of the inclusion period (Vf) or death. Cox regression analysis was used to identify prognostic factors for time to progression or death using univariate and multivariate analysis (forward stepwise). All variables for which *p* < 0.10 were included in the Cox multivariate analysis. The incidence of all, severe, and mild adverse events was analyzed. The analysis was performed using R Commander.

## 3. Results

### 3.1. Clinical and Epidemiological Characteristics

The study population comprised 204 patients with ILD-SAI, of whom 123 (60.3%) had RA, 58 (28.4%) had SS, and 23 (11.3%) had IM. 

After a median (IQR) follow-up of 56 (29.8–93.3) months in patients with ILD, 175/204 (85.7%) remained in follow-up, that is, 615.5 patient-years for RA, 462.4 patient-years for SS, and 255.9 patient-years for IM. A total of 29/204 patients (14.2%) died. The mean (SD) survival from diagnosis to death was 242.3 (13.3) months. No significant differences in survival were recorded between RA, SS, and IM (mean [95% CI], 184.4 months [17.1] vs. 200.3 months [20.6] vs. 267.0 (16.4) months; *p* = 0.469 [log-rank test]).

The main baseline characteristics for the overall sample and the 3 subgroups are shown in Table 1. The three groups differed in terms of epidemiologic, clinical, and laboratory characteristics, as well as in the treatment they received. Women accounted for more than half of the patients with ILD-SAI (66%). The mean age was 65 years, and almost half of the patients had been smokers or were smokers at inclusion. Compared with the other groups, patients with RA were more equally balanced in terms of sex, had a higher mean age, and were more frequently smokers or former smokers. Similarly, the three groups differed with respect to the specificity of antibodies and treatment. More than 80% of patients with RA were RF- or ACPA-positive, whereas those with SS more frequently had a positive anti-scl70 titer (48%), followed by a positive anticentromere titer (34%). The most frequent antibodies in IM were anti-Jo (26%) and PL-7 (21%). 

All patients were receiving treatment for ILD at inclusion. More than half were taking csDMARDs (63%), almost 40% were taking bDMARDs, 35% were taking immunosuppressants, and 2% were taking antifibrotics. Almost 70% of patients were receiving corticosteroids at a median dose of 5 mg/d. Differences in treatment were observed between the subgroups, with mainly csDMARDs and bDMARDs in patients with RA and immunosuppressants in those with IM and SS. Rituximab was the most common bDMARD (37/204 patients [18.1%]), although it was used more frequently in patients with SS (24%) and IM (17%) than in those with RA (15%). Patients with SS took corticosteroids less frequently and at lower doses than the other subgroups. 

The most common radiological pattern was NSIP (99/204 [48.5%]) followed by UIP (91/2014 [44.6%]), fibrotic NSIP (7/204 [3.4%]), and other types of ILD (7/204 patients [3.4%]). By patient subgroup, NSIP was more common in patients with IM (87%), and the UIP pattern was the most common in RA (62%). While the NSIP pattern predominated in patients with SS (72%), almost 30% had UIP or fibrotic NSIP (*p* < 0.001). Chest X-ray and high-resolution CT positive for different interstitial lung disease patterns in patients with ILD-SAI are shown in Figure 1.

### 3.2. Clinical Course

A total of 132/204 patients (64.7%) developed an infection during follow-up. Most affected the respiratory tract (56.4%), and 75/204 patients (36.8%) were admitted to hospital at least once. The most common reason for hospitalization was respiratory infection (23.5%), followed by progression of lung disease (10.3%). No significant differences in respiratory infection were recorded between the different subgroups, whereas skin infections were more common in patients with SS (20.6%) and IM (17.3%) than in those with RA (7.3%) (*p* = 0.032). Hospitalization was more common in RA (44.7%), followed by SS (27.6%) and IM (17.4%) (*p* = 0.010) (Table 2). Twenty-nine patients died (14.2%): 13 from progression of lung disease and superinfection, 10 from rapidly progressing ILD, and 6 from tumors and progression of lung disease.

As for the main lung outcome at the end of follow-up (Table 3), most patients improved or stabilized and 36% worsened or died. By subgroup, progression or mortality was more frequent in patients with RA (48/123 [39%]) than in those with SS (21/58 [36%]) and IM (4/23 [17%]) (*p* = 0.016). Mean survival until progression of lung disease or death was 136.3 (11.0) months, with a median value that was greater for IM than for SS and RA (median [95% CI], 171.7 [107.8–210.3] vs. 159.0 [137.9–174.7] vs. 111.3 [65.0–127.3] months; *p* = 0.017 [log-rank]).

As shown in Table 3, mean PFT values worsened at the end of follow-up compared with baseline. By subgroup, FVC, FEV1, and DLCO decreased significantly at the end of follow-up in patients with RA, as did FVC and DLCO in patients with SS (Figure 2). Furthermore, as seen in the table, no significant differences in PFT values at diagnosis were recorded between the subgroups. However, at the end of follow-up, FVC and DLCO values were lower in patients with RA and SS than in those with IM (*p* = 0.036 and *p* = 0.026, respectively).

HRCT revealed disease progression in 72/204 patients (35.3%), whereas 44/204 (21.5%) met the criteria for progression of ILD and 29/204 (14.2%) died. Progression by HRCT was more pronounced in patients with RA and SS (*p* = 0.002). Analysis of lung disease by subgroup revealed more frequent progression and death in 48/123 (39%), followed by 21/58 (36%) in those with SS and 4/23 (17%) in those with IM (*p* = 0.016). Mean survival until progression and death was 136.3 (11.0) months, with a higher median value for IM than for SS and RA (median [95% CI], 171.7 [107.8–210.3] vs. 159.0 [137.9–174.7] vs. 111.3 [65.0–127.3] months; *p* = 0.017 [log-rank]).

### 3.3. Factors Associated with Progression of Lung Disease and Death in ILD-SAI

Table 4 shows the results of the Cox multivariate analysis (dependent variable: progression or death) performed in 204 patients with ILD-SAI over a median 56.5 (29.8–93.3) months. The outcome was progression or death in 73/206 patients. The multivariate analysis revealed that smoking, UIP pattern, FVC < 80% at initiation of follow-up, and diagnosis of RA vs. SS and IM were associated with a higher probability of progression of lung disease and death.

## 4. Discussion

We found that after 5 years’ progression of the three types of ILD-SAI, ILD stabilized or improved in almost two-thirds of patients. These results are similar or even superior to those observed in other cohorts of patients with ILD-SAI [5,40,41,42]. This improvement is probably associated with increased use of specific immunosuppressants and biologics [43,44]. Furthermore, the predominance of patients with RA (followed by SS and IM) reflects the prevalence reported for each of these diseases elsewhere [45]. 

The poorer course of lung disease in patients with RA than in those with SS and IM (*p* = 0.016) is consistent with reports of better survival for IM and SS than other SAI in patients with ILD [40,46,47], especially RA [45,48]. The particularly poor result for patients with RA is probably due to specific epidemiological, clinical, and radiological characteristics of this disease. Our multivariate analysis revealed that, together with the diagnosis of RA, the UIP pattern, baseline FVC < 80%, and current or previous smoking were independently associated with poor pulmonary outcomes. In contrast with observations in patients with SS and IM, in whom the NSIP pattern (considered more inflammatory and less fibrosing) is predominant, the incidence of UIP, the most fibrotic subtype, is higher in HRCT [49]. The inflammatory forms generally respond better to immunosuppression than the fibrotic forms [42], as reflected in the different survival rates. For example, 5-year survival was 36% in patients with ILD-RA and the UIP pattern and 94% in those with the NSIP pattern [25]. Similarly, 5-year survival in patients with SS and IM and a mainly NSIP pattern ranges from 60% to 85% [2]. Nevertheless, the HRCT pattern is not the only important one, and other authors have reported that progression of ILD-SS may not depend only on histopathology, since the 5- and 10-year survival rates in patients with a UIP or NSIP pattern in the biopsy were 82–90% and 29–69%, respectively [50].

Another factor associated with poorer outcomes in ILD was smoking, which was more prevalent in patients with RA. While some studies did not show active smoking to be a risk factor for mortality or progression of ILD-SAI [15,51], others that evaluated smoking as active or previous (as in our study) did in fact find smoking to be a predictor of progression of lung disease [5]. The third factor associated with more frequent progression and mortality in our study was greater baseline impairment of FVC. Given that this factor was also observed in RA [2,11,15,51], identification of these three factors could enable more intensive therapy or earlier referral for assessment of lung transplantation.

Furthermore, as observed elsewhere, other factors may play a role and were identified only in our bivariate analysis. Thus, we observed that patients with IM and ILD responded particularly well to mycophenolate or rituximab, with fewer decreases in PFT values and more frequent stabilization of HRCT than in other ILD-SAI groups [52]. 

Disease-specific prognostic factors also included the greater presence of men and older patients in RA than in the other groups [53], in contrast with findings for ILD-SS and ILD-IM [54]. Another difference was the variation in antibody specificity and the type of treatment received. More than 80% of patients with RA had positive RA or ACPA values, whereas those with SS and IM mainly had greater percentages of anti-Scl70 and antisynthetase antibodies, respectively. This observation is consistent with data reported elsewhere [2,4].

Our study is subject to a series of limitations. First, its retrospective design meant that there was no information on progression during the different periods of the course of the disease. However, clinical, laboratory, and HRCT and PFT data were available for all patients at onset and at inclusion, thus making it possible to determine progression at the end of follow-up. Furthermore, while we considered various SAIs with different pathogenic mechanisms (thus hampering identification of subtle predictors in prognosis), the main objective of our study was to know the clinical and epidemiological differences in and progression of lung disease for each of the three subgroups studied. Additionally, it is noteworthy that although the imbalance between RA and SS/IM could have hampered the analysis, the large sample size revealed sufficient differences between the subgroups. Despite these limitations, our study is one of the few that identifies epidemiologic and clinical differences, as well as differences in how progress of the various types of ILD-SAI affects the lungs, since this will provide a better picture of the progress of each group of patients and make it possible to identify patients who may require more intensive therapy or earlier referral for evaluation of lung transplant.

## 5. Conclusions

In conclusion, lung function stabilized in more than half of the patients with ILD-SAI after a median of 5 years of follow-up, although somewhat more than one-third progressed rapidly and died. Our data indicated that smoking, UIP pattern, and greater decrease in FVC at disease onset predict progression and death in patients with ILD-SAI. Furthermore, progression and mortality were worse for patients with RA than for those with SS or IM. The ability to predict which patients will progress could facilitate prognosis and early treatment. The morbidity and mortality associated with ILD-SAI make it necessary to perform more studies so that we can identify more risk factors and design models for predicting progression and mortality in ILD-SAI.

## Figures and Tables

**Figure 1 diagnostics-11-01794-f001:**
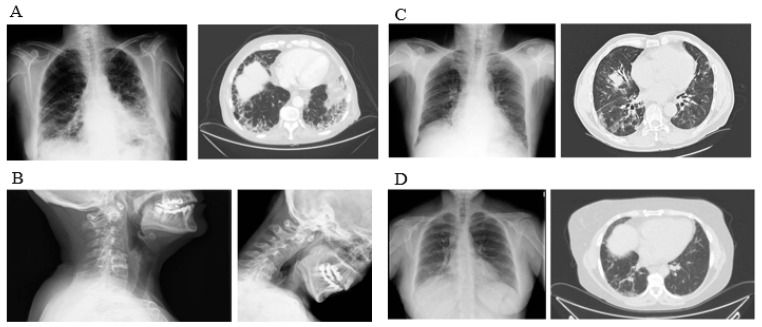
Chest X-ray and high-resolution CT positive for different interstitial lung disease patterns in patients with associated systemic autoimmune disease: (**A**) Pattern usual interstitial pneumonia (UIP) in a patient with rheumatoid arthritis. (**B**) Atlantoaxial subluxation in a patient with interstitial lung disease and rheumatoid arthritis. (**C**) Pattern nonspecific interstitial pneumonia (NSIP) in inflammatory myopathy. (**D**) Pattern nonspecific interstitial pneumonia (NSIP) in systemic sclerosis.

**Figure 2 diagnostics-11-01794-f002:**
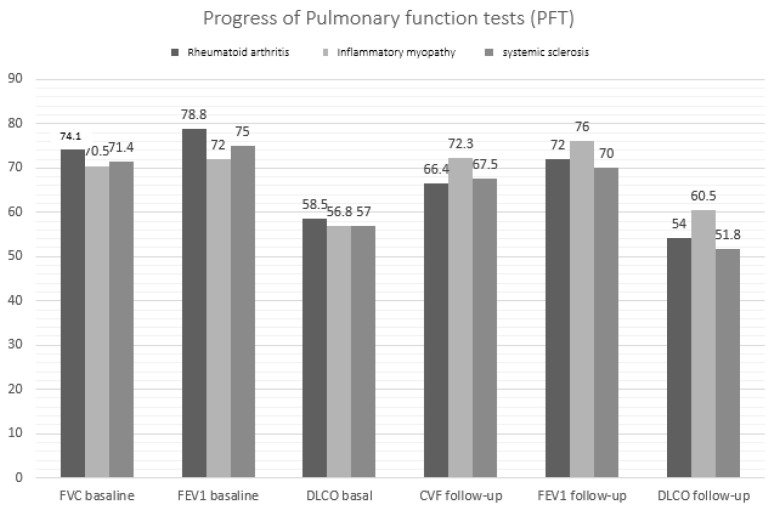
Changes in pulmonary function values by ILD-SAI subgroup.

**Table 1 diagnostics-11-01794-t001:** Clinical-epidemiologic characteristics.

	Total (n = 204)	RA (n = 123)	IM (n = 23)	SS (n = 58)	*p* Value
Epidemiological characteristics					
Female sex, n (%)	136 (66.7)	67 (54.5)	18 (78.3)	51 (87.9)	<0.001
Caucasian race, n (%)	198 (97.1)	119 (96.7)	23 (100)	56 (96.6)	0.673
Age, years, mean (SD)	65.3 (12.6)	69.2 (9.6)	55.5 (18.5)	61.0 (12.2)	<0.001
Clinical and laboratory characteristics					
Smoking					0.002
Nonsmokers, n (%)	112 (54.9)	54 (43.9)	19 (82.6)	39 (67.2)	
Smokers, n (%)	36 (17.6)	28 (22.8)	1 (4.3)	7 (12.1)	
Exsmokers, n (%)	56 (27.5)	41 (33.3)	3 (13.0)	12 (20.7)	
Time with SAI, months, median (IQR)	124.0 (57.4–209.7)	146.3 (68.2–218.1)	51.7 (25.1–157.5)	119.3 (56.6–206.1)	0.037
Time with ILD, months, median (IQR)	56.5 (29.8–93.3)	51.8 (28.5–85.9)	33.9 (25.1–80.5)	65.5 (40.3–140.1)	0.059
RF-positive (>10), n (%)	122 (59.8)	119 (96.7)	1 (4.3)	2 (3.4)	<0.001
ACPA (>20), n (%)	107 (52.5)	107 (87.0)	0 (0.0)	0 (0.0)	<0.001
ANA-positive, n (%)	104 (51.0)	30 (24.6)	19 (82.6)	55 (94.8)	<0.001
Anti-SCL70, n (%)	28 (13.8)	0 (0.0)	0 (0.0)	28 (48.3)	<0.001
Anticentromere, n (5)	20 (9.8)	0 (0.0)	0 (0.0)	20 (34.4)	<0.001
PM-SCL, n (%)	3 (1.5)	0 (0.0)	0 (0.0)	3 (5.1)	0.033
RNP, n (%)	2 (1.0)	0 (0.0)	0 (0.0)	2 (3.4)	0.079
Anti-Ku, n (%)	1 (0.5)	0 (0.0)	0 (0.0)	2 (3.4)	0.079
RNA-polymerase 3, n (%)	1 (0.5)	0 (0.0)	0 (0.0)	1 (1.7)	0.282
Anti-Jo, n (%)	6 (2.9)	0 (0.0)	6 (26.1)	0 (0.0)	<0.001
Anti-PL7, n (%)	5 (2.5)	0 (0.0)	5 (21.7)	0 (0.0)	<0.001
Anti-EJ, n (%)	1 (0.5)	0 (0.0)	1 (4.3)	0 (0.0)	0.019
Anti-NMDA, n (%)	3 (1.5)	0 (0.0)	3 (13.0)	0 (0.0)	<0.001
Anti-TIF, n (%)	2 (1.0)	0 (0.0)	2 (8.7)	0 (0.0)	<0.001
Anti-SRP, n (%)	1 (0.5)	0 (0.0)	1 (4.3)	0 (0.0)	0.019
Secondary Sjögren syndrome, n (%)	33 (16.2)	18 (14.6)	6 (26.1)	9 (15.5)	0.387
Treatment					
Conventional synthetic DMARDs, n (%)	129 (63.5)	105 (85.4)	12 (52.2)	12 (20.7)	<0.001
Methotrexate, n (%)	62 (30.4)	53 (43.1)	4 (17.4)	5 (8.6)	<0.001
Leflunomide, n (%)	32 (15.7)	31 (25.2)	0 (0.0)	1 (1.7)	<0.001
Sulfasalazine, n (%)	9 (4.4)	9 (7.3)	0 (0.0)	0 (0.0)	<0.001
Hydroxychloroquine, n (%)	36 (17.7)	23 (18.7)	8 (34.7)	5 (8.6)	0.016
Biologic DMARDs, n (%)	78 (38.2)	57 (46.3)	6 (26.1)	15 (25.9)	0.013
Anti-TNF, n (%)	14 (6.9)	14 (11.4)	0 (0.0)	0 (0.0)	0.343
Tocilizumab, n (%)	8 (6.9)	5 (4.1)	1 (4.3)	2 (3.4)	0.343
Abatacept, n (%)	19 (9.3)	19 (15.4)	0 (0.0)	0 (0.0)	0.001
Rituximab, n (%)	37 (18.1)	19 (15.4)	4 (17.4)	14 (24.1)	0.365
Immunosuppressants	75 (36.9)	15 (12.2)	20 (87.0)	40 (68.9)	<0.001
Mycophenolate, n (%)	58 (28.5)	10 (8.1)	13 (56.5)	35 (60.3)	<0.001
Azathioprine, n (%)	14 (6.9)	5 (4.1)	6 (26.1)	3 (5.1)	0.009
Cyclophosphamide, n (%)	3 (1.5)	0 (0.0)	1 (4.3)	2 (3.4)	0.095
Antifibrotic drugs, nintedanib n (%)	4 (2.0)	2 (1.6)	0 (0.0)	2 (3.4)	0.549
Corticosteroids, n (%)	134 (66.0)	88 (71.5)	19 (82.6)	27 (46.6)	0.004
Dose of corticosteroids, median (IQR)	5.0 (0.0–7.5)	5.0 (0.0–7.5)	5.0 (5.0–10.0)	2.5 (0.0–5.0)	0.003

Abbreviations. RA: rheumatoid arthritis; IM: inflammatory myopathy; SS: systemic sclerosis; ILD: interstitial lung disease; RF: rheumatoid factor; ACPA: anticitrullinated protein antibodies; ANA: antinuclear antibodies, RNP: anti-U1RNP antibodies; DMARD: disease-modifying antirheumatic drug; SD: standard deviation. Statistical tests used: Pearson χ2, ANOVA, and Kruskal-Wallis.

**Table 2 diagnostics-11-01794-t002:** Clinical events.

	Total (n = 204)	RA (n = 123)	IM (n = 23)	SS (n = 58)	*p* Value
Infections, n (%)	132 (64.7)	82 (66.7)	15 (65.2)	35 (60.3)	0.666
Respiratory infection, n (%)	116 (56.9)	76 (61.8)	12 (52.2)	28 (48.3)	0.205
Other infections, n (%)	59 (28.9)	34 (27.6)	7 (30.4)	18 (31.0)	0.883
Cold sore (herpes), n (%)	10 (4.9)	7 (5.6)	1 (4.3)	2 (3.4)	0.242
Cutaneous, n (%)	25 (12.2)	9 (7.3)	4 (17.3)	12 (20.6)	0.032
Urinary infection, n (%)	28 (13.7)	19 (15.4)	3 (13.0)	6 (10.3)	0.460
Hospitalization, n (%)	75 (36.8)	55 (44.7)	4 (17.4)	16 (27.6)	0.010
Reason for hospitalization					0.032
Respiratory infection, n (%)	48 (23.5)	37 (30.1)	4 (17.4)	7 (12.1)	
Progression of ILD, n (%)	21 (10.3)	15 (12.2)	0 (0.0)	6 (10.3)	
Other causes, n (%)	6 (2.9)	3 (2.4)	0 (0.0)	3 (5.2)	
Mortality, n (%)	29 (14.2)	19 (15.4)	1 (4.3)	9 (15.5)	0.355

Abbreviations. ILD: interstitial lung disease; SAI: systemic autoimmune diseases; RA: rheumatoid arthritis; IM: inflammatory myopathy; SS: systemic sclerosis.

**Table 3 diagnostics-11-01794-t003:** Results of pulmonary function testing.

		Total (n = 204)	RA (n = 123)	IM (n = 23)	SS (n = 58)	*p* Value
Outcomes clinical course **						0.016
Improvement, n (%)	Final	33 (16.1)	14 (11.4)	9 (39.1)	10 (17.2)	
Stabilization, n (%)	Final	98 (48.0)	61 (49.5)	10 (43.5)	27 (46.6)	
Worsening, n (%)	Final	44 (21.5)	29 (23.5)	3 (13.0)	12 (20.7)	
Death, n (%)	Final	29 (14.2)	19 (15.4)	1 (4.3)	9 (15.5)	
Pulmonary function testing						
FVC, mean (SD)	Baseline	72.9 (16.6)	74.1 (15.8)	70.5 (15.8)	71.4 (18.5)	0.465
	Final	68.2 (16.2) *	66.4 (21.4) *	72.3 (15.8)	67.5 (22.8) *	0.036
FVC < 80%, n (%)	Baseline	119 (58.3)	73 (60.9)	11 (49.8)	35 (60.3)	0.772
	Final	134 (65.6) *	83 (67.4) *	13 (56.5)	38 (65.5) *	0.045
FEV_1_, mean (SD)	Baseline	77.7 (16.2)	78.8 (16.2)	72.0 (15.5)	75.5 (15.9)	0.168
	Final	72.9 (19.6) *	71.9 (20.4) *	76.0 (17.1)	70.0 (18.5)	0.240
DLCO-SB, mean (SD)	Baseline	57.8 (15.0)	58.5 (15.0)	56.8 (14.3)	56.9 (15.4)	0.775
	Final	53.8 (16.4) *	54.1 (16.2) *	60.5 (15.1)	51.8 (16.5) *	0.026
HRCT pattern						
Radiologic pattern						<0.001
UIP, n (%)	Baseline	86 (42.1)	74 (60.1)	2 (8.7)	10 (17.2)	
	Final	91 (44.6)	77 (62.6)	2 (8.7)	15 (25.8)	
NSIP, n (%)	Baseline	101 (49.5)	35 (28.5)	20 (86.9)	46 (79.3)	
	Final	99 (48.5)	33 (26.8)	20 (86.9)	42 (72.4)	
Fibrotic NSIP, n (%)	Baseline	10 (4.9)	7 (5.6)	1 (4.3)	2 (3.4)	
	Final	7 (3.4)	6 (4.8)	1 (4.3)	1 (3.4)	
Other types, n (%)	Baseline	7 (3.4)	7 (5.6)	0 (0.0)	0 (0.0)	
	Final	7 (3.4)	7 (5.6)	0 (0.0)	0 (0.0)	
Progression by HRCT						0.002
Progression, n (%)	Final	72 (35.3)	48 (39.0)	4 (17.4)	20 (34.5)	
Stabilization, n (%)	Final	93 (45.6)	60 (48.8)	7 (30.4)	26 (44.8)	
Improvement, n (%)	Final	39 (19.1)	15 (12.2)	12 (52.2)	12 (20.7)	

Abbreviations. RA: rheumatoid arthritis; IM: inflammatory myopathy; SS: systemic sclerosis; ILD: interstitial lung disease; FVC: forced vital capacity; FEV1: forced expiratory volume in the first second; DLCO: diffusing capacity of the lung for carbon monoxide; UIP: usual interstitial pneumonia; NINE: nonspecific interstitial pneumonia; HCRT: high-resolution computed tomography; * *p* < 0.005 final vs. baseline; ** Total progression of lung disease: based on HRCT and PFT (FVC and DLCO). Statistical tests used: Pearson χ2, ANOVA, Kruskal-Wallis, paired t, and Wilcoxon.

**Table 4 diagnostics-11-01794-t004:** Cox regression model (adjusted for progression of ILD).

Variable	Univariate HR (95% CI)	Multivariate HR (95% CI)	*p* Value
Age, years	1.025 (1.00–1.05)		
Male sex	1.172 (0.64–2.14)		
Current or previous smoking	2.419 (1.48–3.94)	2.799 (1.64–4.75)	0.010
UIP pattern	2.317 (1.44–3.71)	1.787 (1.06–2.99)	0.028
Positive ACPA or scl70 values	2.046 (1.31–3.19)		
Baseline FVC < 80%	2.840 (1.47–5.48)	2.348 (1.40–3.92)	0.015
Baseline DLCO-SB < 80%	3.696 (0.86–9.12)		
Corticosteroids	1.434 (0.76–2.08)		
csDMARDs	1.080 (0.595–1.96)		
Immunosuppressants	0.937 (0.52–1.66)		
bDMARDs	0.744 (0.40–1.35)		
SAI subtype	2.101 (1.20–3.66)	1.901 (1.32–2.73)	0.027

Abbreviations. RA: rheumatoid arthritis; ILD: inflammatory lung disease; UIP: usual interstitial pattern; ACPA: anticitrullinated peptide antibody; FVC: forced vital capacity; DLCO-SB: diffusing capacity of the lung for carbon monoxide (single breath); csDMARD: conventional synthetic disease-modifying antirheumatic drug; bDMARD: biologic disease-modifying antirheumatic drug; SAI: systemic autoimmune disease. Independent variables: Sex, age, smoking history, radiological pattern (UIP/nonspecific interstitial pneumonia), positive values for ACPA and anti-SCL70 antibodies, baseline FVC, baseline DLCO-SB, corticosteroids, csDMARDs (methotrexate, leflunomide, hydroxychloroquine, sulfasalazine), immunosuppressants (azathioprine, mycophenolate), bDMARDs (anti-TNF, anti-IL6, abatacept, and rituximab), SAI subtype (inflammatory myopathy, systemic sclerosis, and rheumatoid arthritis).

## Data Availability

Data presented in this study are available on request from the corresponding author.

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
