# Peer review of "Characteristics and Predictors of Progression Interstitial Lung Disease in Rheumatoid Arthritis Compared with Other Autoimmune Disease: A Retrospective Cohort Study"

_diagnostics, 2021, doi:10.3390/diagnostics11101794_

Round 1

Reviewer 1 Report

Nicely organized pathological consequences in patients with ILD/SAI.  

My acceptance was based on the followin POSITIVE points:

1. Early recognition of patients at high risk for being subjected to serious morbid complications, specifically who require lung transplant.

 2. The diversity of clinical presentation (clinical characterstics) in different forms of autoimmune diseases, needs prompt knowledge and experience

3. We cannot consider the current  findings re (ILD-SAI )as novel. But neveretheless, it can be taken as a useful educational and alarming  tool for both rheumatologists and pulmonologists .

4. It might be useful if authors can add axial pulmonary CT scan of their patients with SAI . Also if possible to add some radiographs to show invasion of the adjacent skletal structures such as the clavicles and associated rib lesions. 

5.  It could be useful if authors show full dynamic lateral cervical spine RADIOGRAPH  to assess C1/2 instability because of degenerative process in patients with autoimmune, especially rheumatoid arthritis. 

Author Response

Comments for the reviewers

We would like to thank the editor for considering our work for publication in “Diagnostics” and the reviewers for their comments, which have helped to improve the quality of our manuscript.

Below, we provide a point-by-point reply to the comments.

Reviewer #1: 

Comments and Suggestions for Authors

Nicely organized pathological consequences in patients with ILD/SAI.  

My acceptance was based on the followin POSITIVE points:

  1. Early recognition of patients at high risk for being subjected to serious morbid complications, specifically who require lung transplant.

Reply: We agree with the reviewer and we have added this comment in the discussion:

Pag 10, line 293-300: “Another factor associated with poorer outcomes in ILD was smoking, which was more prevalent in patients with RA. While some studies did not show active smoking to be a risk factor for mortality or progression of ILD-SAI (15, 51), others that evaluated smoking as active or previous (as in our study) did in fact find smoking to be a predictor of progression of lung disease (5). The third factor associated with more frequent progression and mortality in our study was greater baseline impairment of FVC. Given that this factor was also observed in RA (2, 15, 51, 52), identification of these 3 factors could enable more intensive therapy or earlier referral for assessment of lung transplantation”

  1. The diversity of clinical presentation (clinical characterstics) in different forms of autoimmune diseases, needs prompt knowledge and experience

Reply: We agree with the reviewer and we have added this comment in the discussion:

Pag 10, line 305-311: “Disease-specific prognostic factors also included the greater presence of men and older patients in RA than in the other groups (54), in contrast with findings for ILD-SS and ILD-IM (55). Another difference was the variation in antibody specificity and the type of treatment received. More than 80% of patients with RA had positive RA or ACPA values, whereas those with SS and IM mainly had greater percentages of anti-Scl70 and antisyn-thetase antibodies, respectively. This observation is consistent with data reported else-where (2, 4).”

  1. We cannot consider the current  findings re (ILD-SAI )as novel. But neveretheless, it can be taken as a useful educational and alarming  tool for both rheumatologists and pulmonologists.

Reply: We agree with the reviewer, appreciate your comment and hope that it can be taken as a useful educational and alarming tool for both rheumatologists and pulmonologists.

  1. It might be useful if authors can add axial pulmonary CT scan of their patients with SAI . Also if possible to add some radiographs to show invasion of the adjacent skletal structures such as the clavicles and associated rib lesions. 

Reply: In line with the reviewer's suggestion, we have added chest X-ray and high-resolution CT positive for different interstitial lung disease patterns in patients with associated systemic autoimmune disease (ILD-SAI) in figure 1.

Pag 4, line 191-196: “By patient subgroup, NSIP was more common in patients with IM (87%), and the UIP pattern was the most common in RA (62%). While the NSIP pattern predominated in patients with SS (72%), almost 30% had UIP or fibrotic NSIP (p<0.001). Chest X-ray and high-resolution CT positive for different interstitial lung disease patterns in patients with ILD-SAI are shown in Figure 1.”

Figure 1: Chest X-ray and High-resolution CT positive for differents interstitial lung disease patterns in patients with associated systemic autoimmune disease A) pattern usual interstitial pneumonia (UIP) in a patient with rheumatoid arthritis. B) Atlantoaxial subluxation in a patient with interstitial lung disease and rheumatoid arthritis. C) pattern nonspecific interstitial pneumonia (NSIP) in inflammatory myopathy. D) pattern nonspecific interstitial pneumonia (NSIP) in systemic sclerosis.

  1. It could be useful if authors show full dynamic lateral cervical spine RADIOGRAPH  to assess C1/2 instability because of degenerative process in patients with autoimmune, especially rheumatoid arthritis.

Reply: According to the reviewer, we have added radiograph with atlantoaxial subluxation in a patient with interstitial lung disease and rheumatoid arthritis (figura 1,B)

Figure 1: Chest X-ray and High-resolution CT positive for differents interstitial lung disease patterns in patients with associated systemic autoimmune disease A) pattern usual interstitial pneumonia (UIP) in a patient with rheumatoid arthritis. B) Atlantoaxial subluxation in a patient with interstitial lung disease and rheumatoid arthritis. C) pattern nonspecific interstitial pneumonia (NSIP) in inflammatory myopathy. D) pattern nonspecific interstitial pneumonia (NSIP) in systemic sclerosis.

Reviewer #2: 

Comments and Suggestions for Authors

This is a retrospective study that describes the characteristics and progression of interstitial lung disease in patients with associated systemic autoimmune disease and tries to identify factors associated with progression and mortality. 

  1. The article is well structured however in the discussions section the authors should underline further the novelty of their research and position in the current literature.

Reply: In line with the reviewer's suggestion, we have added the novelty of our Research and position in the current literature in the discussion:

Pag 10, line 324-329: De-spite these limitations, our study is one of the few that identify epidemiologic and clinical differences, as well as differences in how progress of the various types of ILD-SAI affects the lungs, since this will provide a better picture of the progress of each group of patients and make it possible to identify patients who may require more intensive therapy or earlier referral for evaluation of lung transplant.

  1. The references are not in the correct format in the text or in the reference section. Please bring up to date.

Reply: We have checked all the references. We have ordered the bibliography 41 and have reviewed the rest. We appreciate his comment.

Reviewer 2 Report

This is a retrospective study that describes the characteristics and progression of interstitial lung disease in patients with associated systemic autoimmune disease and tries to identify factors associated with progression and mortality. 

The article is well structured however in the discussions section the authors should underline further the novelty of their research and position in the current literature.

The references are not in the correct format in the text or in the reference section. Please bring up to date.

Author Response

(The authors gave the same response as above.)
